# The Interaction between Psychological Stress and Iron Status on Early-Life Neurodevelopmental Outcomes

**DOI:** 10.3390/nu15173798

**Published:** 2023-08-30

**Authors:** Brie M. Reid, Michael K. Georgieff

**Affiliations:** 1Department of Psychiatry and Human Behavior, Warren Alpert Medical School, Brown University, Providence, RI 02912, USA; 2Center for Behavioral and Preventive Medicine, The Miriam Hospital, Providence, RI 02906, USA; 3Division of Neonatology, Department of Pediatrics, University of Minnesota Medical School, Minneapolis, MN 55455, USA; georg001@umn.edu

**Keywords:** psychological stress, pregnancy, infancy, iron status, inflammation, neurodevelopment

## Abstract

This review presents evidence from animal and human studies demonstrating the possible connection and significant impact of poor iron status and psychological distress on neurocognitive development during pregnancy and the neonatal period, with implications for long-term cognition. Stress and iron deficiency are independently prevalent and thus are frequently comorbid. While iron deficiency and early-life stress independently contribute to long-term neurodevelopmental alterations, their combined effects remain underexplored. Psychological stress responses may engage similar pathways as infectious stress, which alters fundamental iron metabolism processes and cause functional tissue-level iron deficiency. Psychological stress, analogous to but to a lesser degree than infectious stress, activates the hypothalamic–pituitary–adrenocortical (HPA) axis and increases proinflammatory cytokines. Chronic or severe stress is associated with dysregulated HPA axis functioning and a proinflammatory state. This dysregulation may disrupt iron absorption and utilization, likely mediated by the IL-6 activation of hepcidin, a molecule that impedes iron absorption and redistributes total body iron. This narrative review highlights suggestive studies investigating the relationship between psychological stress and iron status and outlines hypothesized mechanistic pathways connecting psychological stress exposure and iron metabolism. We examine findings regarding the overlapping impacts of early stress exposure to iron deficiency and children’s neurocognitive development. We propose that studying the influence of psychological stress on iron metabolism is crucial for comprehending neurocognitive development in children exposed to prenatal and early postnatal stressors and for children at risk of early iron insufficiency. We recommend future directions for dual-exposure studies exploring iron as a potential mediating pathway between early stress and offspring neurodevelopment, offering opportunities for targeted interventions.

## 1. Introduction

The micronutrient iron is critical for the developing brain [1]. Iron deficiency’s impact on the developing brain has been reviewed extensively and includes acute effects on neurodevelopment alongside increased risk to lifelong mental health [2]. In total, 20–30% of all pregnant women have iron deficiency anemia (IDA), and 40% of children under five years of age are iron-deficient (ID; [3]). Though intervention through iron supplementation can support neurodevelopment, there is growing concern that oral iron treatment might not reverse all neurodevelopmental risks of ID, especially in the context of systemic inflammation [4,5]. While intervention research has focused on reducing inflammation through reducing pathways to infectious stress (e.g., integrated water, sanitation, and hygiene interventions), the contributions of how non-infectious psychological stress have yet to be rigorously evaluated. Non-infectious psychological stress responses utilize many of the same pathways as infectious stress to alter basic processes of nutrient metabolism, including absorption and prioritization [6]. Like infectious stress, psychological stress activates and, if chronic, dysregulates the hypothalamic–pituitary–adrenocortical (HPA) axis and increases proinflammatory cytokines [7]. The dysregulated neuroendocrine pathways that arise from chronic psychological stress can potentially disrupt iron absorption, distribution, and utilization, most likely through the activation of hepcidin by the proinflammatory cytokine IL-6, which is elevated in both infectious stress and psychological stress. Even with adequate iron intake, the activation of hepcidin by stress may disrupt maternal, fetal, and infant iron transport and distribution. Furthermore, iron deficiency and psychological stress independently affect similar neurodevelopmental outcomes. If maternal and infant stress and neuroendocrine dysregulation alter iron prioritization and loading through hepcidin, the resolution would suggest a non-nutritional—i.e., reduction in stress—instead of a nutritional solution—i.e., providing more iron.

This paper serves as a narrative review of the current literature to outline the effect of psychological stress pre- and neonatally and the risk it may carry for offspring iron status and neurodevelopment. Our central hypothesis is that chronic psychological stress, through the dysregulation of the HPA axis and proinflammatory mechanisms, negatively impacts maternal, fetal, and infant iron status, resulting in risk to early neurodevelopment. As the populations most at risk of iron deficiency are also those experiencing high burdens of psychological stress, we will first review psychological stress and the growing literature that examines associations between psychological stress and iron status. Second, we will review our hypothesized biological pathways between psychological stress and iron absorption and use, focusing on the HPA axis, inflammation, and hepcidin. Then, we will highlight the shared neurodevelopmental outcomes of pre- and early-life stress and iron deficiency. Finally, we will outline the current outstanding questions and future directions in this field of research. Our goal with this review is to create a framework for future studies to elucidate the effects of prenatal and neonatal stress and HPA mechanisms on the risk of functional iron deficiency in the mother and offspring iron outcomes. Ultimately, future research will help to identify outcome-specific biological mechanisms, markers, and modifiable risk factors as intervention targets.

## 2. Psychological Stress and Iron Status

### 2.1. Psychological Stress—Definitions and Prevalence

For our review, we focus on psychological stressors as either severely acute or chronic exposures that would threaten human survival throughout evolution [8] and require the individual’s stress-mediating systems to make physiological compensations to support allostasis [9]. This definition includes real and interpreted physical danger from another individual, whether through physical assault, abuse, or neglect, and social conflict and isolation [8]. For pregnant people, infants, and young children, stress exposure during these times is associated with altered offspring development [10]. Between 65% to 70% of pregnant women in the United States report at least one stressful life event during pregnancy [11,12], with higher rates for non-Hispanic Black women and American Indian/Alaska Native women [13]. Additionally, one in five women experiences multiple stressful life events while pregnant. Given the prevalence of maternal stress, low to moderate stress levels are likely adaptive. High or chronic levels likely carry risks. Around the world, ~300 million children aged two to four years experience stress in the form of physical and psychological violence [14], and infants under two are especially vulnerable [15]. One in four children under five lives with a mother who is a victim of intimate partner violence [16]. These stress exposures are especially relevant to iron deficiency, as populations and areas with a high prevalence of iron deficiency are also areas where many pregnant people and young children are exposed to chronic stressors such as food insecurity, poverty, and conflict [17,18,19,20].

### 2.2. Studies on Stress and Iron Status

#### 2.2.1. Prenatal Stress and Offspring Iron Status

Exposure to stress, including measures of objective external stressors and self-reports of psychological distress, has been directly related to offspring iron status. A study of mothers living in an area under rocket attack in the first trimester of pregnancy had offspring with lower cord blood ferritin than the offspring of mothers who were pregnant after the rocket attacks ended [21]. Higher levels of maternal self-report subjective stress were also associated with lower cord blood ferritin concentration, especially for the group of mothers exposed to rocket-attack stress [21]. A study of 493 mother–infant dyads found that mothers with higher levels of prenatal perceived stress, violence exposure, and anxiety symptoms had infants with lower cord blood ferritin concentrations [22]. The impact of stress during pregnancy extends into infancy. Pregnant women who experienced higher stress levels during pregnancy had one-year-old infants at a higher risk of having low plasma ferritin [23]. Additionally, a study of pregnant rhesus macaques found that monkeys exposed to a laboratory stressor during pregnancy, particularly during the second trimester, had offspring with lower iron levels as they grew compared to unexposed offspring [24]. In multiple studies, evidence is accumulating to support how psychological stress prenatally places offspring at risk of worse iron status at birth and in infancy.

#### 2.2.2. Postnatal Stress and Iron Status

Preclinical studies show evidence for changes in iron metabolism following postnatal stress exposure. Studies with rodents have shown that exposure to psychological stress for 1–2 weeks leads to decreased serum iron levels, increased hepatic iron content, and reduced protein expression of the iron exporter ferroportin (FPN) in the duodenum and liver [25,26,27,28]. In adult rodents, rodents exposed to psychological stress exhibit decreases in serum iron, hemoglobin, ferritin, and erythropoietin concentrations [26]. Psychological stress also decreases iron absorption and impairs iron transporter expression in the small intestine of adult rats [29]. Exposure to acute and chronic stressors in rodents reduces whole-blood iron concentration [25]. In contrast, the administration of dexamethasone for three weeks increases serum iron levels and reduces liver content and the expression of transferrin receptor 1 (TFR1) [30], while repeat restraint stress upregulates liver TFR2 expression and leads to liver iron accumulation [31]. As these studies were conducted using adult animals, it is unclear how these findings would relate to humans, especially prenatally and during infancy. Only a few studies have investigated the impact of stress on iron levels in humans. For instance, a study conducted on Navy SEAL trainees revealed that a week of psychological stress disrupted their iron levels [32]. Additionally, Chilean children exposed to accumulated family-level stressors during infancy were found to have poorer iron levels and were more likely to be diagnosed with IDA [33]. However, one drawback of the current research in this field is that there is a lack of studies conducted on infants, children, or animals in their early developmental stages.

## 3. Shared Neurodevelopmental Outcomes of Pre-Natal and Early-Life Stress and Iron Deficiency

The high demand for iron in infancy coincides with the rapid growth and development of brain structure and functions that require iron, including the hippocampus, cortical regions, neuronal and glial energy metabolism, myelin synthesis, and neurotransmission [34]. Iron is also essential for synthesizing serotonin, norepinephrine, and dopamine neurotransmitters [34]. Offspring of mothers with iron deficiency anemia (IDA) are at high risk for low birth weight, prematurity, small for gestational age, and poor neurodevelopment [2]. IDA in pregnancy can negatively impact the iron endowment of the neonate, which may cause irreversible harm to neurodevelopment [4,17]. Postnatally, iron-sufficient newborn infants are especially at risk of becoming iron-deficient (ID) between 6 and 12 months of age, when prenatal iron stores become depleted [2,5]. Iron deficiency is particularly damaging during the critical 6–24-month postnatal window of rapid brain development, as several key brain areas require iron for normal development [1].

Animal studies show associations between altered brain metabolism, myelination [35,36,37,38,39], neurotransmitter function [40], and early-life iron deficiency. Early ID is also associated with alterations to the developing hippocampus [41], with pervasive and long-lasting iron deficiency-induced metabolic [42] and dendritic structure changes [43]. Neurophysiologic studies of the effects of iron deficiency have found differences in the speed of neural transmission in the auditory system, recognition memory, longer auditory brainstem response latencies, and longer visual evoked potential latencies (for review, see [34]). Infants at high risk for ID show poorer recognition memory, possibly due to iron’s effects on the hippocampus and central nervous system. Iron deficiency during infancy is associated with children’s socioemotional and behavioral problems and lower cognitive abilities [40]. Thus, inadequate iron can negatively impact neurodevelopment across several domains and in different brain regions.

The consequences of fetal iron under-loading include abnormal acute neonatal brain function and long-term associations with mental health and psychopathology. Fetal iron under-loading during gestation carries the risk of earlier onset of postnatal iron deficiency, as the newborn is born with lower stores of iron which are depleted earlier. In turn, postnatal iron deficiency anemia is associated with both acute and long-term neurobehavioral abnormalities. Insufficient iron in the fetal period is associated with brain function that includes worse recognition memory [44,45], slower speed of neural processing [46], and poor bonding and maternal interaction [47]. Long-term, fetal iron deficiency is associated with an increased risk of autism spectrum disorder diagnosis (first trimester ID) [48], schizophrenia (second trimester ID) [49], and other neurocognitive impacts, including impaired memory performance (third trimester ID) [50,51]. Postnatal iron deficiency is associated with motor dysfunction [52], social-emotional behavior [52], and an increased risk of depression and anxiety in adulthood [53]. Research in rodent models shows that early iron deficiency causes specific anatomical, physiological, and molecular brain changes in different regions [42]. Iron deficiency during prenatal and neonatal stages impacts the developing hippocampus, a crucial structure for learning, memory, and the neuroendocrine stress response [43,54].

Similar to insufficient fetal iron loading, prenatal stress impairs learning [55], increases anxiety and depressive behaviors [56] (reviewed in [10]), and affects dopaminergic and hippocampal development [57,58]. Exposure to stress during prenatal development is linked to cortical thinning and reduced cognitive functioning in offspring [59]. Prenatal stress exposure is also associated with emotional reactivity in preschool-age offspring [60]. Over the lifespan, fetal exposure to maternal stress has been associated with an increased risk for autism spectrum disorder (ASD), schizophrenia [61], anxiety, depression, and ADHD [62,63]. Postnatal stress is also associated with neurobehavioral outcomes similar to iron deficiency, including impacts on learning, socioemotional development, and increased risk for mood disorders and psychopathology. Severe stress limited to infancy in the form of early institutional care impairs attention regulation and executive functioning (EF) (reviewed in [64,65,66]), thought to be due to changes in prefrontal-striatal and anterior cingulate circuitry [67]. Rodents exposed to early-life stress display alterations in synaptic signaling and epigenetics in the hippocampus and amygdala, linked to increased anxiety and depressive-like behaviors (reviewed in [68,69,70,71,72,73,74,75]). Rodents subjected to postnatal stress experience reduced dendritic arborization in both the PFC and hippocampus (reviewed in [68,76,77]). Studies have shown a connection between changes in hippocampal synaptic plasticity caused by stress and decreased spatial memory learning in rodents [68,78,79]. The neurodevelopmental outcomes of fetal and early postnatal iron insufficiency share similarities with those arising from fetal and early exposure to psychological stress, summarized in Table 1. Still, to our knowledge, no study has elucidated both exposures’ combined contributions to neurodevelopment.

## 4. Hypothesized Biological Pathways between Psychological Stress and Iron Status

Biological and sociological reasons exist for the co-occurrence of psychological stress and ID [82]. For instance, young children experiencing stress in the form of poverty may be at higher risk of iron deficiency due to food insecurity or diets low in iron (e.g., [83]). Maternal ID also commonly occurs among communities affected by poverty, which contributes to fetal iron under-loading. The rate of ID in pregnancy in the US is 42% [84], with higher rates in less-resourced populations [3]. Behaviorally, stress has been found to change how individuals eat, and pregnant women and children under stress may eat with a preference for high-fat, high-sugar foods that are likely lower in iron (reviewed in [85,86]). While a lack of access to or reduced preference for iron-rich foods is an important factor to consider, more relevant to this review are the biological changes associated with psychological stress that may impact iron status even in the context of adequate intake. The stress-induced biological changes we will focus on are the HPA axis, inflammation, hepcidin, actions on protein, and other potential mechanisms of note, outlined in Figure 1.

### 4.1. Stress and the HPA Axis

The hypothalamic–pituitary–adrenal (HPA) axis is a complex physiological system that plays a crucial role in our response to stress. The HPA axis involves multiple systems, including the autonomic, neuroendocrine, metabolic, and immune systems, which work in tandem to help manage stress and maintain homeostasis. One of the key functions of the HPA axis is the release of glucocorticoids (i.e., cortisol) in response to potential threats or danger [88]. The ability to mount a stress response and return to baseline is essential for survival: the mature HPA axis ideally responds to stress with a peak in cortisol and a swift return to baseline [89]. The HPA axis also exhibits a diurnal rhythm, with a rise in cortisol levels at waking and a peak 30–45 min after waking, followed by a decline across the day and a nadir in cortisol levels around bedtime [90,91,92,93]. Maintaining this diurnal rhythm is essential for mental and physical health, and chronic stress has been found to disturb the diurnal cortisol rhythm across multiple populations [90].

Current research suggests that experiencing moderate stress levels can benefit our overall functioning [94]. However, prolonged exposure to high stress levels can negatively impact physical and mental health [95,96,97]. The relationship between cortisol and optimal functioning follows an inverted U-shaped curve: cortisol that is too low and too high either in response to a stressor or in the diurnal rhythm tends to impair functioning [10]. Extended periods of low cortisol levels can impair restorative functions if the mineralocorticoid receptors are not fully occupied. On the other hand, extended periods of high cortisol levels can result in an overabundance of stress- or energy-depleting functions mediated by mineralocorticoid receptors due to a shift in the ratio of glucocorticoid receptor to mineralocorticoid receptor occupation (reviewed in [89]).

How chronic stress impacts HPA axis functioning depends on which stage of development the organism experiences the stress in (for review, see [98]). In pregnancy, severe acute or chronic stress can impact the mother’s HPA axis and the offspring’s developing brain and HPA axis [99] (for review, see [86,100]). Some maternal cortisol does reach the fetal compartment, and maternal stress may decrease placental 11β-HSD-2 activity, allowing more maternal cortisol to enter fetal circulation as it crosses the placenta ([101], for review, see [102]). At birth, humans have a functioning HPA axis that exhibits an immature diurnal rhythm and an HPA axis that can react to environmental changes, arousal levels, and distress [103,104]. This axis continues to mature across development. In this way, chronic or severe stress exposure during the prenatal or postnatal period can impact the HPA axis and subsequently influence iron metabolism. Preclinical work has found a connection between stress hormones and iron metabolism in the brain: in vitro corticosterone application dysregulates iron metabolism in hippocampal neurons [105]. To our knowledge, the only study in humans that has examined this connection provides preliminary evidence that maternal HPA axis regulation in response to an acute stressor is associated with iron status in the third trimester of pregnancy [106]. We hypothesize that the primary actor in this cascade involves inflammatory mechanisms, which we will review in the next section.

### 4.2. Stress and Inflammation

Decades of research have established connections between stress and immune function, including inflammation ([107], for reviews, see [8,108,109]). There are multiple pathways by which stress-mediating systems communicate with the peripheral immune system and can induce a proinflammatory state that can influence iron status. These pathways include the HPA axis, the sympathetic nervous system (SNS), the vagus nerve, and meningeal lymphatic vessels. Psychological stress triggers an inflammatory response centrally and peripherally (e.g., [110,111]). Increases in levels of circulating proinflammatory cytokines and expression of proinflammatory genes typically follow exposure to an acute stressor (minutes to hours) [112,113]. Similar to the increase in innate immune activity and proinflammatory cytokines during infectious stress, this increase in a proinflammatory profile is considered adaptive in the short term [108] but can be maladaptive—especially for iron status—in the long term.

Chronic stress is also associated with increased inflammation, decreases in antiviral immunity, changes in adaptive immunity [107,109], and increased expression of inflammation-related genes, which have been hypothesized to shift an organism into a proinflammatory phenotype [114]. Early life stress increases the signaling and upregulation of genes associated with inflammation in primates, children, and adults [115,116,117,118]. Stress exposure during childhood is also associated with increased circulating markers of inflammation in pediatric populations [119] and adult populations with a history of childhood stress exposure [120]. Adolescents who experienced higher levels of childhood stress have exhibited increased inflammatory gene expression [121], which aligns with similar findings found in adults (as reviewed in [108]). Stress early in life may result in an epigenetic modification in the GR gene responsible for regulating the body’s inflammatory response, most notably in the neural transcriptome of the hypothalamus and amygdala [122]. Additionally, stress can epigenetically reprogram immune cells and impact their production from the bone marrow, resulting in an imbalance in the body’s immune response [123,124]. In pregnant individuals, studies of mothers with high levels of psychological stress and low social support find associations between stress and elevated proinflammatory cytokines levels across gestation [86,125]. How stress impacts the developing immune system and leads to a proinflammatory phenotype in human neonates and infants is still an active area of research [126]. Nevertheless, the evidence suggests that non-infectious psychological stress may operate along similar pathways of infection and inflammation [127], leading to the risk of IDA even in the context of adequate iron intake.

### 4.3. Stress, Inflammation, and Hepcidin

Hepcidin may be the key factor in the relationship between psychological stress, inflammation, and iron status. Hepcidin, its receptor, and iron transporter ferroportin work in concert to control iron’s dietary absorption, storage, and tissue distribution. Hepcidin is upregulated in response to inflammatory states to decrease iron availability and control infection. Hepcidin and ferroportin expression are modulated during infection and inflammation—and potentially psychological stress—to reduce iron availability. Iron supply for red blood cell precursors is also restricted, contributing to the anemia associated with infections and inflammatory conditions [128]. Hepcidin signaling causes functional iron deficiency (redistribution of iron into the reticuloendothelial system, away from the red cells) on top of iron deficiency from other causes, such as inadequate dietary iron, or pregnancy and early infancy’s physiologic demands for iron. In chronic stress, the hepcidin-mediated reduction in gut absorption may also worsen total body iron deficiency. IL-6 is a key proinflammatory cytokine that rises after an acute stressor, is found in studies of chronic stress [108,112], and is concurrently the cytokine required for inducing hepcidin during inflammation [129]. Thus, increased hepcidin activity would increase the risk of iron deficiency when increased inflammation is due to psychological stress. Preliminary evidence supports the stress-inflammation-hepcidin pathway. In a study of adult rats, psychological stress from a communication box paradigm induced hypoferremia through the IL-6–hepcidin axis [27]. Psychological stress increased IL-6 and hepcidin expression, and the changes were reversed by IL-6 monoclonal antibody injection [27]. In humans, a study of 370 adolescent boys from Hyderabad, India, found that higher levels of self-reported life event stressors were associated with elevated IL-6 and hepcidin concentration [130]. To our knowledge, no studies have investigated this link in pregnant or infant populations.

### 4.4. Other Mechanisms

Prenatally, the placenta plays a crucial role in regulating fetal development and the intrauterine environment. Maternal stress can impact the placental methylome, which may affect fetal outcomes and increase the risk of fetal iron-deficiency anemia [131,132]. However, the placenta’s role in fetal IDA risk in the context of maternal stress and maternal IDA remains understudied. There could be a connection between psychological stress, iron levels, mitochondrial functioning, and neurodevelopment. In non-pregnancy data, both stress and iron deficiency can affect the functioning of mitochondria, which are responsible for regulating and signaling metabolism in cells and have been implicated in developing psychopathology [133]. Iron is essential for neuronal energy metabolism [134]. For instance, early-life iron deficiency can disrupt the size, motility, and energy capacity of mitochondria in developing hippocampal neurons [135], as well as impair mitochondrial energetics and the transcriptional regulation of mitochondrial quality control genes in adult animals that were previously iron-deficient [136]. In a rodent model of prenatal stress, male offspring displayed depressive-like symptoms associated with a reduction in PGC-1α protein, a regulator of mitochondrial biogenesis, in the frontal cortex and hippocampus [137]. A human study also showed that prenatal stress can affect placental mitochondrial DNA gene expression, and increased expression of MT-ND2 was subsequently linked to infant temperament [138]. Furthermore, a recent study in mice found that chronic social stress can disrupt iron metabolism and enhance hepatic mitochondrial function and ATP production [139].

Another possible connection between psychological stress and iron status involves the disruption of protein metabolism. In cases of infectious stress, steroids and proinflammatory cytokines can disrupt protein metabolism, causing the body to go into a catabolic state to produce fuel for fight-or-flight responses. This means that proteins are broken down for gluconeogenesis instead of being used for building tissue, including the brain’s structural proteins like dendrites and synapses, as well as other soluble proteins like neurotransmitters and growth factors (which can be affected by stress/sepsis and reduce IGF-1 synthesis). Additionally, proteins are necessary to transport iron around the body (as hemoglobin or cytochromes to generate ATP). Linear growth stunting is one illustrative example of how stress and inflammation can suppress protein synthesis. Postnatal linear growth stunting in infants and toddlers is a likely somatic instantiation of inflammation (e.g., [140]). In a parallel population, in preterm infants in the NICU, stunting is due to a lack of nutrition (protein-energy) and the amount of steroid, antibiotic, and inflammation exposure [141]. Much of these effects are mediated through effects on mammalian Target of Rapamycin (mTOR) pathway signaling [142], which integrates nutrient availability and growth factor status to regulate neuronal growth and differentiation. Therefore, the connection between psychological stress and iron may also be linked to stress and inflammation’s effects on protein. Additional pathways between stress and iron metabolism are likely and warrant further research.

## 5. Conclusions and Future Directions

Chronic psychological stress and HPA axis dysregulation in pregnant and non-pregnant individuals are associated with increased levels of circulating proinflammatory cytokines IL-6 and CRP. IL-6 directly regulates hepcidin synthesis. The neurodevelopmental consequences of early stress and iron deficiency may reflect mirrored impacts or interactions between the neuroendocrine response to stress and iron metabolism. Therefore, a future area of research that needs attention is the dysregulated neuroendocrine pathways that arise from psychological stress that may disrupt iron absorption and utilization even in the context of adequate intake [143,144].

Several outstanding questions will need to be addressed in future research. In studies of adult rodents, it is unclear why and how stress-related imbalances in iron homeostasis are linked to the type, duration, and severity of stressors. Prenatally, psychological stress in monkeys early in pregnancy (second trimester) affected iron status in infant offspring [24], and dietary intervention with moderate iron supplementation during pregnancy did not prevent the offspring from developing iron deficiency [145]. The second trimester is also highlighted in the association between prenatal iron deficiency and the risk of offspring schizophrenia in epidemiological work in humans [49]. Future research must address the timing, type, and duration of psychological stress and the subsequent impacts on iron metabolism.

Furthermore, despite suggestive connections between psychological stress and iron status in preclinical animal work (e.g., [24]) and cohorts of pregnant women [21,22,23], no study has examined how hypothalamic–pituitary–adrenocortical (HPA) axis dysregulation arising from chronic psychological stress affects maternal risk for iron deficiency and offspring iron status. In studies of pregnant rhesus macaques, the magnitude of the maternal cortisol response over pregnancy was not associated with offspring iron status [24]. Though one preliminary study in humans found evidence for associations between maternal cortisol response to an acute stressor and iron status in the third trimester [106], the precise relationship between the HPA axis and iron regulation in pregnancy and infancy remains an open area of research. There are also still questions about whether psychological stress causes an increase in proinflammatory cytokines that leads to changes in hepcidin and protein regulation, which affects iron status. Additionally, it is unclear if there is a threshold for the relationship between HPA axis regulation, inflammation, and iron metabolism changes or if it is a continuous relationship, and if so, what the biomarker for that threshold is. Further research is needed to fully understand the complicated relationship between psychological stress, inflammation, and iron metabolism.

Evaluating maternal chronic stress as a novel treatment mechanism could prevent maternal and offspring IDA and subsequent health problems. This line of research has the potential to change the clinical infrastructure required to solve prenatal maternal iron deficiency and risk to offspring in high-stress populations, requiring a different allocation of resources. Studying the effects of maternal psychological stress, neuroendocrine regulation, and inflammatory biomarkers on iron status could help identify novel intervention targets focused on psychological stress for preventing IDA and subsequent neurodevelopmental sequelae. Overall, further research on the relationship between maternal chronic stress and iron deficiency has the potential to benefit the health and well-being of both pregnant people and their children.

## Figures and Tables

**Figure 1 nutrients-15-03798-f001:**
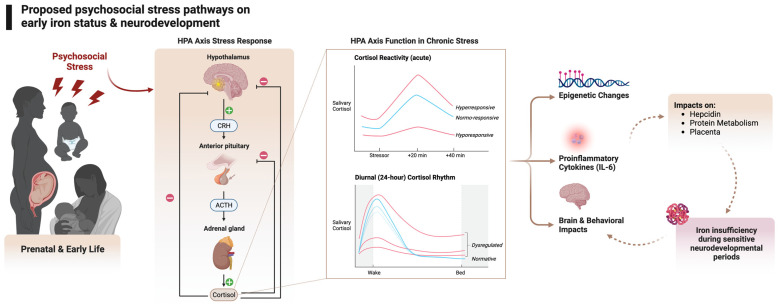
Exposure to psychological stress triggers the hypothalamic pituitary adrenocortical (HPA axis) response. The HPA axis response to a stressor begins with the release of corticotropin-releasing hormone (CRH) and arginine vasopressin (AVP) by neurons in the medial parvocellular region of the paraventricular nucleus of the hypothalamus. These hormones stimulate the pituitary gland to secrete adrenocorticotropic hormone (ACTH), which triggers the adrenal cortex to produce glucocorticoids. These hormones bind to corticosteroid receptors (the glucocorticoid receptor (GR) and the mineralocorticoid receptor (MR)) throughout the brain and regulate gene expression, leading to various physiological and psychological effects [10]. The key to the HPA axis is feedback loops to maintain homeostasis. Once the perceived stressor has subsided, the feedback loops at multiple levels, including the hypothalamus, hippocampus, and frontal cortex, shut down the HPA axis and return the organism to homeostasis [10]. If a stressor is chronic, dysregulated cortisol output is seen in acute stress responses and the diurnal pattern of cortisol. These cortisol patterns are associated with epigenetic changes, increases in proinflammatory cytokines, and brain and behavior changes. We hypothesize that increased IL-6 from psychological stress impacts key pathways between iron metabolism and brain development. Figure created with Biorender.com [87].

**Table 1 nutrients-15-03798-t001:** Shared neurodevelopmental consequences of insufficient iron and stress early in life.

Insufficient Iron during Gestation and in Infancy	Maternal Stress Exposure and HPA Activation in Pregnancy, Infancy, Toddlerhood
Abnormal acute brain function	
Poor recognition memory * [44,45] Slower speed of neural processing * [46] Poor bonding and maternal interaction * [47] Impacts on hippocampus [41,43,54]; metabolic [42] and dendritic structure changes *^†^ [43] Altered brain metabolism, myelination [35,36,37,38,39], neurotransmitter function *^†^ [40]	↑ cortical thinning and decrements in cognitive functioning * [59] Alterations in synaptic signaling and epigenetics in the hippocampus and amygdala, linked to **↑** anxiety and depressive behaviors ^†^ [69,70,71,72,73,74,75] ↓ Dendritic arborization in PFC, hippocampus ^†^ [76,77] Changes in hippocampal synaptic plasticity, **↓** spatial memory learning ^†^ [68,78,79]
Acute and long-term neurobehavioral abnormalities	
Motor Dysfunction ^†^ [52] Socio-Affective ^†^ [52] Neurocognitive, including **↓** memory performance (3rd trimester ID) * [50,51]	Impaired learning * [55] Impaired attention regulation and EF ^†^ [64,65,66] ↑ Anxiety and depressive behaviors * [56] ↑ Emotional reactivity in preschool-age offspring * [80]
Long-term mental health abnormalities	
↑ Risk of ASD (1st trimester ID) * [48] ↑ Risk of schizophrenia (2nd trimester ID) * [49]	↑ Risk for ASD schizophrenia [61] ↑ Risk of anxiety, *^†^ depression, *^†^ ADHD * [53,62,63,81]

* = prenatal exposure, ^†^ = postnatal exposure, ID = iron deficiency, ↓ = decreased, ↑ = increased, ADHD = attention deficit hyperactivity disorder, PFC = prefrontal cortex, ASD = autism spectrum disorder, and EF = executive functioning.

## Data Availability

Not applicable.

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
