# Peer review of "The Interaction between Psychological Stress and Iron Status on Early-Life Neurodevelopmental Outcomes"

_nutrients, 2023, doi:10.3390/nu15173798_

Round 1
Reviewer 1 Report
This is a well written review article on the important topic of iron deficiency in early life development. The authors present evidence for the perspective that early life stress exacerbates risk of iron deficiency with implications for offspring neurocognitive development.
Minor comments to improve the readability and flow of the manuscript:
1. The authors switch between the terms "psychosocial" and "psychological" in various places throughout, particularly in sections 1 and 2. Suggest using consistent terminology or else defining what is meant by either term to justify the alternative use in some instances. In line 284, the term "social stress" is used - this should either be psychosocial or psychological stress.
2. Line 164-168: this sentence is quite long and wordy. Suggest breaking into two sentences as follows: "Fetal iron under-loading during gestation carries the risk of earlier onset of postnatal iron deficiency, as the newborn is born with lower stores of iron which are depleted earlier. In turn, postnatal iron deficiency anemia is associated with with both acute and long-term neurobehavioral abnormalities."
3. Line 181: missing an open parentheses symbol "("
4. Line 191: missing a close parentheses symbol ")"
5. Line 200-201: It seems incorrect to state that no studies have examined the independent contributions of stress or iron deficiency on neurodevelopmental outcomes, since this entire section summarizes evidence for the independent effects of each factor. It would be more correct to only emphasize that the combined effects have yet to be considered.
6. Table 1: The term "neurodevelopmental" should be included in the table title. Suggest revising the first column header to either "Insufficient fetal iron & iron deficiency in infancy" or "Insufficient iron during gestation and in infancy". The second column header reads awkwardly; recommend removing "Similarities to" and the remainder will make sense by itself. Also suggest removing bullet points within the table.
7. Line 210: Assuming this sentence is meant to relate to the prior one about poverty, suggest editing to read "Maternal ID also commonly occurs among communities affected by poverty, which contributes to fetal iron under-loading."
8. Figure 1. The line labels within the cortisol graphs have such tiny font they are difficult to read. Please enlarge font size
9. Line 320-323: the second part of this sentence (referring to prenatal/infant physiological demands on iron) feels out of context given that this paragraph is not specific to early life physiology. Consider making the statement more general then giving the example of prenatal/infant ID. E.g. "...on top of iron deficiency from other causes, such as inadequate dietary iron, or pregnancy and early infancy’s physiologic demands for iron."
10. Line 354-355: Need to make clear that this sentence refers to non-pregnancy data and then bring it into context of placental function to flow with the rest of the paragraph.
11. Line 377: Insert "axis" after HPA
12. The title and opening of the abstract refer to "neurocognitive development" but the majority of the paper uses the broader term "neurodevelopment", which seems more apt. Consider editing the title to say "early-life neurodevelopmental outcomes".
Author Response
Thank you so much for your review - we are grateful for your kind words and your detailed edits to improve the readability and flow of the manuscript!
Reviewer 1
This is a well written review article on the important topic of iron deficiency in early life development. The authors present evidence for the perspective that early life stress exacerbates risk of iron deficiency with implications for offspring neurocognitive development.
Minor comments to improve the readability and flow of the manuscript:
- The authors switch between the terms "psychosocial" and "psychological" in various places throughout, particularly in sections 1 and 2. Suggest using consistent terminology or else defining what is meant by either term to justify the alternative use in some instances. In line 284, the term "social stress" is used - this should either be psychosocial or psychological stress.
Thank you for this suggestion - we have replaced all instances with social stress and psychosocial with psychological, except in instances where we are referring to social stressors in laboratory animals, where we retain the term social stress.
- Line 164-168: this sentence is quite long and wordy. Suggest breaking into two sentences as follows: "Fetal iron under-loading during gestation carries the risk of earlier onset of postnatal iron deficiency, as the newborn is born with lower stores of iron which are depleted earlier. In turn, postnatal iron deficiency anemia is associated with both acute and long-term neurobehavioral abnormalities."
Thank you for this suggested edit - we have revised the sentence as recommended.
- Line 181: missing an open parentheses symbol "("
Parentheses symbol added
- Line 191: missing a close parentheses symbol ")"
Parentheses symbol added
- Line 200-201: It seems incorrect to state that no studies have examined the independent contributions of stress or iron deficiency on neurodevelopmental outcomes, since this entire section summarizes evidence for the independent effects of each factor. It would be more correct to only emphasize that the combined effects have yet to be considered.
Thank you for catching this, we have removed the reference to independent effects.
- Table 1: The term "neurodevelopmental" should be included in the table title. Suggest revising the first column header to either "Insufficient fetal iron & iron deficiency in infancy" or "Insufficient iron during gestation and in infancy". The second column header reads awkwardly; recommend removing "Similarities to" and the remainder will make sense by itself. Also suggest removing bullet points within the table.
We really appreciate these recommendations and agree that they improve the table. We have made all the suggested changes to Table 1
- Line 210: Assuming this sentence is meant to relate to the prior one about poverty, suggest editing to read "Maternal ID also commonly occurs among communities affected by poverty, which contributes to fetal iron under-loading."
We have edited the sentence as recommended, thank you.
- Figure 1. The line labels within the cortisol graphs have such tiny font they are difficult to read. Please enlarge font size
We apologize for the small font size and have updated the figure as recommended.
- Line 320-323: the second part of this sentence (referring to prenatal/infant physiological demands on iron) feels out of context given that this paragraph is not specific to early life physiology. Consider making the statement more general then giving the example of prenatal/infant ID. E.g. "...on top of iron deficiency from other causes, such as inadequate dietary iron, or pregnancy and early infancy’s physiologic demands for iron."
Thank you for this suggestion - we’ve edited the sentence as recommended.
- Line 354-355: Need to make clear that this sentence refers to non-pregnancy data and then bring it into context of placental function to flow with the rest of the paragraph.
We have edited this sentence to the following: “There could be a connection between psychological stress, iron levels, mitochondrial functioning, and neurodevelopment. In non-pregnancy data, both stress and iron deficiency can affect the functioning of mitochondria, which are responsible for regulating and signaling metabolism in cells and have been implicated in developing psychopathology [135].”
Our line numbers do not seem to be lining up with your noted line numbers, so if we misunderstood the line you were referring to, please let us know. Thank you!
- Line 377: Insert "axis" after HPA
Added, thank you for catching this
- The title and opening of the abstract refer to "neurocognitive development" but the majority of the paper uses the broader term "neurodevelopment", which seems more apt. Consider editing the title to say "early-life neurodevelopmental outcomes".
We have edited the title, thank you for the suggestion.
Reviewer 2 Report
A unique review, extremely well written, with excellent scientific and didactic values.
The senior author is a well-known scientist who has been working in the field of iron metabolism for years, and it was my great pleasure to review this manuscript.
It is rare that I do not have major criticisms of a manuscript, but in this case, I believe the paper should be accepted for publication as it is. Each section is carefully prepared and contributes to understanding the relationship between iron metabolism and stress and neurocognitive development.
The manuscript provides much practical information and the literature review is thorough. The figure and the table certainly help in the analysis of the topic undertaken by the authors.
Author Response
Thank you so much for your kind words. We are excited to get this manuscript into the hands of other scientists, and are thrilled that you feel similarly!